# Current Developments of Artificial Intelligence in Digital Pathology and Its Future Clinical Applications in Gastrointestinal Cancers

**DOI:** 10.3390/cancers14153780

**Published:** 2022-08-03

**Authors:** Alex Ngai Nick Wong, Zebang He, Ka Long Leung, Curtis Chun Kit To, Chun Yin Wong, Sze Chuen Cesar Wong, Jung Sun Yoo, Cheong Kin Ronald Chan, Angela Zaneta Chan, Maribel D. Lacambra, Martin Ho Yin Yeung

**Affiliations:** 1Department of Health Technology and Informatics, The Hong Kong Polytechnic University, Kowloon, Hong Kong SAR, China; nn-alex.wong@polyu.edu.hk (A.N.N.W.); zebang.he@connect.polyu.hk (Z.H.); kalong59.leung@connect.polyu.hk (K.L.L.); ivan.cy.wong@connect.polyu.hk (C.Y.W.); cesar.wong@polyu.edu.hk (S.C.C.W.); jungsun.yoo@polyu.edu.hk (J.S.Y.); 2Department of Anatomical and Cellular Pathology, The Chinese University of Hong Kong, Prince of Wales Hospital, Shatin, Hong Kong SAR, China; curtis.to@cuhk.edu.hk (C.C.K.T.); ronaldckchan@cuhk.edu.hk (C.K.R.C.); mabs_md2001@yahoo.com (M.D.L.); 3Department of Anatomical and Cellular Pathology, Prince of Wales Hospital, Shatin, Hong Kong SAR, China; caz129@ha.org.hk

**Keywords:** digital pathology, computational pathology, whole-slide imaging, artificial intelligence, histopathology, gastrointestinal tract, algorithms, machine learning, deep learning, cancer diagnosis

## Abstract

**Simple Summary:**

The rapid development of technology has enabled numerous applications of artificial intelligence (AI), especially in medical science. Histopathological assessment of tissues remains the gold standard for diagnosis of gastrointestinal (GI) cancers for subsequent management. In a traditional anatomical pathology (AP) laboratory, histopathologists are required to manually assess, quantify and classify diseases under a microscope in a semiquantitative or qualitative manner. The conversion of analogue-to-digital pathology is on the rise, conforming with trends toward digitalization. The rapid adoption of digital pathology (DP) is driven by factors such as the worldwide shortage of pathologists and medical technologists, the increasing incidence of cancer and the critical need to improve laboratory efficiency. In this review, we aim to provide a comprehensive summary of algorithms for AI detection and classification of GI cancer. We will also provide critical insight into the application of algorithms for routine care in clinical practice.

**Abstract:**

The implementation of DP will revolutionize current practice by providing pathologists with additional tools and algorithms to improve workflow. Furthermore, DP will open up opportunities for development of AI-based tools for more precise and reproducible diagnosis through computational pathology. One of the key features of AI is its capability to generate perceptions and recognize patterns beyond the human senses. Thus, the incorporation of AI into DP can reveal additional morphological features and information. At the current rate of AI development and adoption of DP, the interest in computational pathology is expected to rise in tandem. There have already been promising developments related to AI-based solutions in prostate cancer detection; however, in the GI tract, development of more sophisticated algorithms is required to facilitate histological assessment of GI specimens for early and accurate diagnosis. In this review, we aim to provide an overview of the current histological practices in AP laboratories with respect to challenges faced in image preprocessing, present the existing AI-based algorithms, discuss their limitations and present clinical insight with respect to the application of AI in early detection and diagnosis of GI cancer.

## 1. Introduction

GI cancer is a collective term encompassing various cancers related to the GI system, which comprises the organs from the oral cavity down to the anal canal. Malignancy can develop in any part of the GI tract, as it is constantly and directly exposed to carcinogens in the environment through the ingestion of food. Given the numerous hotspots for malignant transformation and the constant exposure to carcinogens, GI cancer accounts for 26% of global cancer incidence and 35% of global cancer-related mortality [1]. The recent surge in incidence and mortality are linked to the increasing prevalence of modifiable risk factors, such as sedentary lifestyle, obesity, unhealthy diet and other metabolic abnormalities [2]. Recent analysis revealed a significant increase in GI cancer incidence in young adults aged 25–49 years, alerting the public to emerging medical burdens [3]. The five major GI cancers causing considerable global burdens are oesophageal squamous cell carcinoma, gastric adenocarcinoma, colorectal cancer (CRC), hepatocellular carcinoma and pancreatic cancer. In order to identify and classify GI cancers, histopathological analysis of endoscopic biopsies or resected tumour specimens remains the gold standard for disease diagnosis. Unfortunately, early diagnosis of GI-related cancers is often missed due to a lack of specific symptoms. Medical attention is often sought only when non-specific symptoms become unbearable [4].

In addition to the existing lack of medical professionals, there are various reasons for the growing demand for histopathological diagnosis, including an increase in morbidity within the aging population, the increasing incidence of malignancies among young adults and the increase in cancer screening programs, resulting in an increased number of annual laboratory cases [5,6,7,8]. Furthermore, the complexity of cases, as well arduous criteria for case reporting, constitute burdens to histopathologists and may prolong the turnaround time (TAT) associated with generating reports in AP laboratories [9]. Standardization among AP laboratories is crucial for pathologists to make precise diagnoses. Medical laboratory technologists are not only responsible for preparing tissue samples for diagnoses but also for ensuring that the laboratory performance is satisfactory through quality assurance programs while operating in compliance with laboratory accreditation [10,11]. With the lack of manpower and restricted laboratory budget, the adoption of more advanced technology is imminent with respect to addressing the overwhelming workload in an AP laboratory.

Whole-slide imaging (WSI) refers to the digitization of traditional glass slides into a digital image. Technological advancement has led to the development of high-throughput WSI scanners, which allow for large amounts of slides to be digitized in a short period of time. The digitized image of stained histological specimens can be used for a variety of purposes. Digital pathology (DP) is a term that incorporates the acquisition, management and interpretation of pathology information [12]. Currently, DP is used in both academic and clinical settings for teaching, diagnostic and archival purposes [13,14,15,16]. More importantly, the worldwide adoption of DP is a trend that could help AP laboratories improve productivity and reduce costs while improving patient care. The establishment of DP workflow in an AP laboratory opens opportunities for computational pathology; image analysis techniques, such as quantification and measurement; and the application of AI in computer-aided diagnosis [17]. The potential of AI in a DP workflow may alleviate burdens faced by histopathologists by reducing the time spent on tedious activities, such as counting cells and measuring tumour parameters, and, likewise, aid in standardizing immunohistochemical (IHC) staining for companion diagnostic testing [18,19,20]. AI-assisted endoscopy can also facilitate the detection and characterization of suspicious polyps, as well as the use of skin surface microscopy to classify skin diseases [21], which may allow for early identification and intervention in cancer. The adoption of AI will be of mutual benefit to healthcare institutes, medical practitioners and patients by increasing laboratory throughput, shortening TAT and reducing operating and patient costs.

The potential of AI in DP and GI-related malignancy is encouraging. However, the process from algorithm development to clinical application is complicated by various obstacles. An example of such a problem is the preprocessing of specimen for WSI generation. This could create variability within datasets [22,23], thus increasing the difficulty and cost of model training in a hospital setting. Machine learning [24,25,26,27] and deep learning [28,29,30,31,32,33,34,35,36,37,38,39,40] are common methods adopted for image analysis and both have unique advantages and disadvantages. Currently, there is a variety of existing models that can be selected and applied easily. Using existing models is convenient but may not be suitable for specific tasks, leading to poor performance. The design and features of models should be carefully considered during algorithm design to complete specific tasks to minimize unnecessary computational time while maintaining high output performance. Models with high performance, as measured by area under the receiver operating characteristic (AUROC), may not necessary be clinically applicable [16,22]. It is important to understand the clinical needs and current acceptability of AI as part of the clinical workflow. As the end users of these algorithm are medical professionals, it is vital to discuss and develop a model specific to their needs and workflow, while also balancing with generalizability for use in other hospital or clinical settings [16,41]. There is a real challenge associated with developing clinically applicable AI-based tools to facilitate pathologists in their workflow. In this regard, we will critically review the processes from preparation of tissue samples to current challenges in applying AI in diagnosis of GI cancers. We will also explore how the development and implementation of AI in DP may improve existing practice in screening and diagnosis of GI malignancies.

## 2. Current Histopathology Practices and Opportunities in Digital Pathology

In modern clinical practice, the advancement of technology enables the development of highly efficient, high-throughput and high-resolution (up to 0.23–0.25 μm/pixel) WSI devices that can digitize traditional glass slides into whole-slide images (WSIs) that can be remotely viewed on a display monitor for remote diagnosis of cases [42,43]. This digital process is termed “digital pathology” and has additional benefits compared to traditional pathology, including the ease of quickly and securely sharing pathology WSIs worldwide with other pathologists for second opinions, as well as providing a better view of the tissue with additional annotation and measurement tools while allowing for rapid access to archived cases without loss of quality [44]. Previous literature reports support the safe transition to digital pathology, showing high concordance of diagnosis using digital pathology with WSIs as compared to traditional slides with light microscopy [45,46,47,48,49]. The quality of WSIs is a major factor that affects concordance rates and diagnostic accuracy. In order for high-quality WSIs to be generated, specific tasks must be performed by medical laboratory technologists. The tissues of interest extracted from esophagogastroduodenoscopy or surgery must be preserved in fixatives, such as 10% neutral buffered formalin, to prevent autolysis of biological tissue while maintaining tissue structure and integrity. The tissue is then embedded, sectioned, stained and mounted onto a glass slide, ensuring the quality of the tissue and glass slide will. This includes checking for artefacts on the tissue, including possible scores, tears, floating contamination, thick and thin sections and ensuring that the glass slide is intact and free from dust before and after digitization [50,51] (Figure 1).

Considering the forthcoming transition from traditional pathology to digital pathology, there is a demand for the development of new tools to facilitate the reporting process for pathologists. In recent years, powerful WSI analysis software tools that are user-friendly yet packed with clinically relevant tools have been developed. The majority of such software applications are open-source and freely available; these include ImageJ 1.53s (Madison, United States), QuPath 0.3.0 (Belfast, United Kingdom), SlideRunner 2.2.0 (Erlangen, Germany) and Cytomine (Liège, Belgium) [52,53,54]. Such software applications are capable of handling large WSIs and metadata generated from different hardware brands and contain interactive drawing tools for annotation. They also include features that can perform cellular detection and feature extraction. Furthermore, to complement software development, the cost of hardware required for high-performance computation, including high-speed network infrastructure and data storage, has become increasingly affordable. This has led to increased adoption of digital pathology in major hospitals worldwide. However, more investment is required to expand the roles of digital pathology in most hospitals [12]. With an increasing number of digital pathology centres, the generation of large and high-quality WSI databases will slowly emerge [17,47,55,56]. This growth increases the feasibility of obtaining large datasets and designing algorithms for analysis of WSIs using computer software through a broad range of methods for the study of diseases. This concept is termed computational pathology [57]. The combination of computational pathology with digital pathology opens new opportunities for case diagnosis. Algorithms developed for the early detection of cancer are important to improve patients’ chance of survival [58]. Automated screening of a large number of specimens may provide improved accessibility and make diagnosis and treatment more affordable [59]. Furthermore, the escalating amount of cancer specimens with increasingly complex classification and the urgent need for shorter TAT have resulted in an upsurge in workload in diagnostic pathology services [60]. With these challenges in mind, more research has been focused on the use of AI to perform diagnostic tasks for histopathological examination. Some models have been proposed to use computational methods to triage patients and present urgent cases to pathologists with prioritizations. Despite the excellent results and potential application in histopathological diagnosis prediction using AI [61,62,63], there are still significant challenges to overcome prior to the implementation of these AI-based tools and algorithms. In the following section, we will explore the critical areas that hinder the current development and deployment of AI-based tools in clinical practice.

## 3. Current Challenges of Algorithm Development in Computational Pathology

AI has become a popular tool in the field of medical image processing and analysis [64]. AI is capable of extracting meaningful information from images that cannot be obtained with the naked eye. Machine learning is a subset of AI that utilizes algorithms developed to process these data and perform tasks without explicit programming. Deep learning is a subset of machine learning that uses highly complex structures of algorithms based on the neural network of the human brain [65] (Figure 2). In recent years, AI has become increasingly popular in histological imaging analysis, with clinical applications that include primary tumour detection, grading and subtyping [66,67,68]. Several well-known research challenge competitions, such as CAMELYON16 and CAMELYON17, encourage the development of novel and automated solutions that are clinically applicable to improve diagnostic accuracy and efficiency [69,70]. This is achieved through various combinations of machine learning and deep learning techniques, which will be discussed further. Large annotated datasets are also indispensable for the development of successful deep learning algorithms. Through the adoption of digital pathology for diagnosis, it is possible for pathologists to annotate regions of interest during the case reporting process, which may indirectly facilitate the generation of such valuable datasets for future algorithm development [16].

Furthermore, the inevitable integration of these AI tools is necessary due to the aging population worldwide and the shortage of pathologists [71]. The training of new pathologists requires a long period of time in order to ensure competency [72]. Thus, there is an urgent need to develop clinically applicable AI-based tools to relieve the high workload of pathologists, producing more precise and reproducible diagnoses while reducing the TAT of cases. However, there remain obstacles and challenges that are hinder the development process, which will be discussed below.

### 3.1. Colour Normalization

For routine histopathological diagnosis, haematoxylin and eosin (H&E) staining is the most preferred method for visualizing cellular morphology and tissue structure. Haematoxylin stains cell nuclei a purplish-blue, whereas eosin stains the extracellular matrix and cytoplasm pink. The patterns of coloration play a central role in the differentiation between cells and structures. Colour and staining variations can be affected by, but are not limited to, the thickness of the specimen, non-standardized staining protocols and slide scanner variations [73]. It is important to perform colour normalization to ensure consistency across WSI databases, as it can affect the robustness of deep learning models. Common methods for colour normalization include histogram matching [74,75], colour transfer [76,77,78] and spectral matching [79,80,81]. However, these methods of colour normalization heavily rely on the expertise of pathologists, which makes WSI colour normalization difficult for general researchers with limited knowledge of the colour profile of histological staining. Moreover, this process of manually assessing and adjusting each WSI is impractical. Several algorithms have been proposed by researchers that are capable of performing colour normalization by deep learning using a template glass slide image for reference, such as StainGAN [82] and SA-GAN [83]. These models have shown promising results with respect to ensuring consistent representation of colour and texture [84,85,86]. Nevertheless, future work is needed to explore the clinical performance of AI-based tools developed using such colour normalization systems.

### 3.2. Pathologist Interpretation for Model Training

For histopathologists, understanding normal histology is essential for investigating abnormal cells and tissue. Final interpretation comprises several processes, including visualization, spatial awareness, perception and empirical experience. When developing new algorithms for the classification of diseases, the wide variation in the interpretation or classification of diseases cannot be categorized or easily represented in a cardinal manner. If so, the classification task is oversimplified, leading to a lack of detail. Physiologically, cancers are characterized by an increase in cellular proliferation, invasive growth, evasion of apoptosis, and altered genome and expression [87]. With H&E staining, histopathologists can identify the changes in cellular characteristics by several broad features of malignant cells, which include anaplasia, loss of polarity, hyperchromatism, nuclear pleomorphism and irregularity, abnormal distribution of chromatin, prominent nucleoli and atypical mitotic figures. Diagnostic reports are often used as the ground truth to label datasets. However, pathologists may use descriptive terminology to explain pathology concepts that are difficult to interpret and categorize [88]. Hence, it is important for researchers to obtain diagnostic reports with standardized reporting, including the specimen type, site, histologic type, grading and staging, in accordance with the American Joint Committee on Cancer, which represents different categories of histopathological knowledge under fuzzy ontology [89], and streamline the process of algorithm development to generate more accurate disease classifications [90].

### 3.3. Model Transparency and Interpretability for Deployment of AI-Based Tools in Clinical Practice

The incorporation of digital pathology, let alone computational pathology, in the clinical workflow requires solid validation of WSI analysis. However, various studies have reported that there is high discordance between diagnoses made using WSI and those made with light microscopy in GI malignancies [46,91,92,93]. More evidence is required to demonstrate that diagnoses made through digital pathology are as performant as conventional light microscopy diagnoses to ensure reliable and safe application in digital pathology practice [91]. According to practical recommendations by the Royal College of Pathologists regarding the implementation of digital pathology [94], validation and verification processes should be performed in the image analysis system to demonstrate its clinical utility before integrating into clinical workflows. Thus, it is also of considerable importance for AI-based tools to be transparent and interpretable in order to address possible moral and fairness biases by providing evidence in making a specific decision [95]. However, there is a dilemma when designing new algorithms, as most algorithm developments focus on using sophisticated deep learning and ensemble methods—so-called “black-box” models—to tackle multidimensional problems. On the other hand, much simpler methods, such as linear regression or decision-tree-based algorithms, may not be sophisticated and powerful enough to achieve the desired outcome [96,97,98,99]. Nonetheless, the demand for more explainable AI is on the rise [100]. Developers of new AI-based tools targeting applications in digital pathology should continuously assess the model explainability of their tool to meet the expectations of various stakeholders. They should also consider the ethical concerns and possible regulatory barriers imposed by governments and professional bodies regulating practice [101,102].

## 4. Machine Learning in GI Cancer Diagnosis

Machine learning algorithms constitute a method that requires the input of large datasets to study patterns or relationships to predict outcomes. The most common machine learning algorithms include artificial neural networks, decision tree algorithms, support vector machine, regression analysis, Bayesian networks, etc. In computational pathology, due to the large image size of each WSI (>3 billion pixels, >1 GB), the majority of these algorithms are designed based on manually extracted or handcrafted features from WSIs to improve their performance and minimise the computational time [103]. Such algorithm design processes can be summarised as feature engineering and can be divided in a two-step process. First, the regions of interest related to a specific clinical problem are identified in images. Then, the images are translated into numerical or categorical values using several predictor variables to create an accurate machine learning algorithm [104]. In digital pathology, there are two kinds of hand-crafted features, namely domain-inspired and domain-agnostic features. Domain-inspired features require the intrinsic domain knowledge of pathologists and oncologists, whereas domain-agnostic features include the general computational features used in the machine learning algorithms [15]. In practice, the generation of handcrafted features is often achieved by software, such as wndchrm, which was developed by Shamir et al. [105], and QuPath by Bankhead et al. [53]. These handcrafted, feature-based machine learning algorithms have demonstrated GI cancer classification, detection and prognostication abilities. The main researches related to machine learning models and algorithms in GI cancer diagnosis are presented in Table 1.

Yoshida et al. [24] pioneers in the application of machine learning algorithms to classify the different grades of gastric cancer (GC) using “e-Pathologist” software. Their system performance showed high sensitivity (>89%) and low specificity (<51%). The authors demonstrated that their software, “e-Pathologist”, is not sensitive enough to detect gastric carcinoma using clinical samples. Similar work has been done by Yasuda et al. [25], with “wndchrm” software; their system exhibited improved performance relative to “e-Pathologist” and showed a superior AUC of >0.98. However, with a limited validation dataset used (<100 WSIs), a larger dataset is required to verify the clinical applicability of wndchrm. Furthermore, Cosatto et al. reported the use of a machine learning algorithm integrated with a multi-instance learning framework to identify the presence of GC [26]. Their framework achieved promising performance (AUC: 0.96), and their validation using a large dataset (4168 patients) illustrates the effectiveness of integration of machine learning with a multi-instance learning architecture. Support vector machine is one of the most popular algorithms in machine learning used for classification and prognostication for cancer research. Jiang et al. [106] used support vector machine to predict the prognosis of GC patients and identify patients who might benefit from adjuvant chemotherapy. Sharma et al. [107] also used support vector machine for detection of gastric tumour necrosis. The use of support vector machine will likely remain popular. Its adoption in computational pathology should be further explored.

To achieve shorter computational time and higher accuracy, machine learning is commonly synergised with deep learning in the field of computational pathology [27,108]. Jiang et al. proposed the combination of the InceptionResNetV2 deep learning model and a gradient-boosting decision tree machine classifier for prognosis prediction of stage III colon cancer. The hybrid model calculated a hazard ratio of 10.273 in the prediction of high-risk or low-risk recurrence and a hazard ratio of 5.033 for patients with poor and good prognoses. [27]

Machine learning algorithms are also capable of performing tasks related to the colour of histology images, as the colour of biomarkers is important for colour normalization of various histology images, as well as disease prediction. Kothari et al. [74] used a four-class linear discriminant classifier to develop a new colour segmentation algorithm for standardization of WSI colour to improve machine learning model accuracy in classifying different colour components from histological cancer images. Furthermore, as aggressive tumour progression is associated with high cell proliferation rates, immunostaining of tumour nuclei is performed by pathologists for estimation of tumour growth. Rokshana et al. [109] proposed the use of an IHC colour histogram to perform colour separation for the quantification of the Ki67 proliferation index for breast cancer. Such a colour-thresholding algorithm can assist pathologists in obtaining automatic and robust prognosis predictions for cancer patients.

## 5. Deep Learning in GI Cancer Diagnosis

Deep learning is a subcategory of machine learning and is used to directly study image patterns. It has also gained popularity in the world of computational pathology. Convolution neural network (CNN) is a widely used deep learning algorithm applied in pathological image analysis. It is a specific type of neural network that uses a convolution operation to process each pixel’s data for image recognition and processing through image processing with different convolution and pooling layers [110]. CNN is able to compress and extract problem-specific information directly from images to train algorithms for the classification, segmentation, detection and prognosis of cancers. Prior to model training, image preprocessing is vital, as most WSIs are excessively large (approx. 40,000 × 50,000 pixels), and the input requirement of CNN allows for small images only (≤512 × 512 pixels). Hence, it is normal practice to tile WSIs prior to algorithm training and development. Herein, we will review a selection of deep learning algorithms developed for GI cancer diagnosis. The main researches related to deep learning models and algorithms in GI cancer diagnosis are presented in Table 2.

### 5.1. Fully Supervised Approach

Fully supervised learning is defined by its use of labelled data to predict outcomes. Computational pathology-related research requires large amounts of pathologist-labelled data with pixel-wise annotation for each WSI for accurate model training.

#### 5.1.1. Classification of GI Cancer

Several deep learning models have been proposed to differentiate between benign and malignant tissue for GI cancer classification. Staging of cancers is also an important parameter that affects the patient’s treatment strategy and prognosis. Recently, research has been focused on the development of high-performance deep learning models to classify normal and cancerous cases with grading. For instance, a ResNet-18 model was reported by Su et al., with an F1 score of >0.86 in terms of distinguishing between poorly differentiated adenocarcinoma and well-differentiated adenocarcinoma [28]. DeepLab is another type of CNN architecture utilized by Song et al. to classify different types of GI cancers and showed promising performance [29,30]. In an earlier publication, Song et al. reported that DeepLab v2 achieved a high AUC (0.92) in recognizing different types of colorectal adenomas (tubular, villous, tubulovillous, etc.) [29]. Later, they utilized an updated DeepLab v3 model for diagnosis of GC with clinical applicability. Their model showed outstanding performance, with an AUC > 0.99 on multiple large external validation datasets, including various difficult cases [30].

Despite research reporting a reasonably reliable model, there is a demand for higher accuracy in differentiating between grades of GI cancer. Alternative novel network structures applied in other types of cancer classification can act as a reference for future model training for GI cancers. Graph transformer network is an example that has demonstrated a high accuracy of 0.97 in the grading of lung carcinoma [111]. Moreover, as WSIs are characterized by high morphological heterogeneity in the shape and scale of tissues, simple tiling of the WSI may compromise spatial information, which may be of benefit in accurately predicting outcomes. A deformable conditional random field (DCRF) model was developed to study the interrelationship by obtaining offsets and weights of neighbouring tiled images. Integration with well-established models, such as ResNet-18, ResNet34 and DenseNet, resulted in improvements in classification accuracy of gastric adenocarcinoma and CRC [112].

Moreover, during the peer-review process of this research, the issue of missed diagnosis by pathologists was identified and discussed. We found that two cases of malignancy were initially missed by pathologists in the initial diagnostic report. However, reassessment by a trained DeepLab v3 AI assistance system identified the malignancy and successfully flagged the tumour regions. This demonstrates that diagnoses by pathologists are not 100% accurate, are subject to interobserver variability and are prone to error when pathologists are facing large workloads and stress. Given the compromised accuracy of diagnoses made solely by pathologists, concerns have been raised about the reliability of using pathologists to train deep learning systems. Interestingly, several studies have shown that deep learning algorithms trained by pathologists exceeded the average performance of pathologists [30,113]. This means that AI systems can analyse and integrate training sets with satisfactory performance, even if all of the annotations by pathologists designated for training might not be fully accurate.

DL-based algorithms are also used for the differentiation of molecular subtypes. Sirinukunwattana et al. [114] developed a deep learning model (imCMS) for the differentiation of four consensus molecular subtype groups with distinct clinical behaviours and underpinning biology from standard H&E-stained WSIs. A similar concept was implemented by Popovici et al. for the classification of low-risk subtypes and high-risk subtypes [115].

The recognition of different types of colorectal polyps (hyperplastic, sessile serrated, traditional serrated adenoma, etc.), cancer-associated tissues (lymphocytes, cancer-associated stroma, colorectal adenocarcinoma epithelium, etc.) and normal tissues is critical to determine the risk of CRC and future rates of surveillance for patients. This has been demonstrated by various research groups using ResNet-152 [116], VGG19 [117] and the ensembled ResNet [118]. Among these models, VGG19 [117] showed the highest model accuracy of 0.943, followed by ResNet-152 (0.93) [116] and ensembled ResNet [118] (0.87). VGG19 (143 million parameters) had a much larger network size than that of ResNet-152 (60 million parameters). It is clear that larger and more complex models with more parameters play an important role in boosting model performance. As a comparable example, the Inception V3 model used by Popovici et al. [115] obtained an average accuracy of 0.72, whereas the VGG-F model used by Sirinukunwattana et al. [114] achieved a greater accuracy of 0.84. This can be explained by the fact that VGG-F (61 million parameters) is more complex than Inception V3 (23 million parameters) with respect to model enhancement. Moreover, a large dataset for model training is also essential for improved prediction results. For example, Gupta et al. [119] used the Inception-ResNet-v2 model to obtain superior performance (F-Score, 0.99; AUC, 0.99) in classifying and locating abnormal and normal tissue regions based on a model trained with more than 1,000,000 WSI patches. Despite the relatively low network complexity of the Inception-ResNet-v2 model (56 million parameters), network performance can be compensated with the support of a large amount of training data.

#### 5.1.2. Segmentation in GI Cancer

Determining the spatial location of cancerous regions in WSI is an important step for accurate diagnosis by pathologists using digital pathology. Various networks have been applied to delineate malignant and benign regions, such as adversarial CAC-UNet [31], U-Net-16 [32] and CoUNet [33]. Remarkably, these networks are trained (750 WSIs) and tested (250 WSIs) with the same DigestPath dataset for detection of the location of gastric cancerous lesions. The adversarial CAC-UNet [31] achieved the highest segmentation performance, with a Dice coefficient score of 0.8749 as compared with U-Net-16 [32], with a Dice coefficient score of 0.7789, and CoUNet [33], with a Dice coefficient score of 0.746. U-Net-16 and CoUNet sequentially classified the corresponding colorectal malignant and benign regions successfully (AUC: 1.00 vs. 0.980), demonstrating that pipelines following segmentation and subsequent classification can potentially achieve high performance in the classification of specific histopathological regions. Ensembling different networks together is another method for performing segmentation tasks in CRC. One such example was reported by Mahendra et al. [120], who developed a general framework combining DenseNet-121, Inception-ResNet-V2 and DeeplabV3Plus models to specifically segment malignant and benign regions.

#### 5.1.3. Detection of Prognostic Markers and Prognosis in GI Cancer

Detecting and identifying prognostic markers is important for the development of the most suitable treatment plan for patients. However, histology WSIs, by nature, contain highly heterogenous cell types, inevitably increasing the dataset difficulty and hindering the model development process of new algorithms to predict outcomes with high prognostic value. Despite these challenges, several models have been proposed to aid in the early detection of prognostic markers that may influence disease progression. The presence of goblet cells with overexpression of TFF3 is a key feature of Barrett’s oesophagus. The identification and quantification of these cells may be indicative of intestinal metaplasia of Barrett’s oesophagus. The VGG16 model showed strong adaptability, with an AUC = 0.88 with respect to detection of patients with Barrett’s oesophagus in TFF3-stained biopsies. Moreover, this approach allowed biopsies to be defined into eight classes with varying priority for pathologists to review. This method can be applied to prioritize important cases for early case review in a busy AP laboratory [34].

Infection with *H. pylori* can lead to chronic inflammation, which is a risk factor for GC. However, *H. pylori* is not easily identified, as the H&E section requires examination under high magnification. This significantly increases the cost and TAT of diagnosis. Hence, early identification of *H. pylori* is a valuable prognostic marker that allows for early and cost-effective treatment to eradicate *H. pylori* and prevent progression to GC. Several DL-based algorithms were tested by combining DenseNet-121 and ResNet-18 models [35]. Overall, this ensemble method achieved satisfactory performance, with accuracy >0.90 in detecting *H. pylori* in human gastric WSIs.

Similarly, identifying microsatellite instability (MSI) in patients with CRC can affect the treatment strategy. However, not all laboratories have the resources to perform MSI testing for each patient. The development of algorithms to directly predict MSI from H&E-stained slides may allow for prediction of MSI status without additional genetic or IHC tests. Shufflenet [36] and ResNet18 with a single dataset [121] and ResNet18 [122] with multiple source datasets were trained and tested for their performance. The models were trained with 6406, 429 and 1230 WSIs, achieving AUCs of 0.92, 0.8848 and 0.69–0.84, respectively. Overall, Shufflenet demonstrated higher performance than the ResNet18 models. Despite the higher WSI number of ResNet18 with multiple source data, its performance was lower than that of ResNet18 trained with a smaller and single database. Overall, the intention of developing a deep-learning-based algorithm is to predict MSI status, with the aim of providing immunotherapy to more patients without the additional cost of MSI testing. It is worth noting that large and reliable WSI databases are essential for the development of highly accurate and performant algorisms.

Similar tactics have been applied to identify MSI in GC. Su et al. reported the use of ResNet-18 to detect MSI status in GC [28]. Their model performance showed poor accuracy of 0.7727 in reporting patient-level MSI status. This is limited by the lack of well-defined regions of MSI status at the tile level. Thus, their study reveals the necessity to obtain accurate pixel-level annotated MSI labels from tumour tissue regions for better algorithm development for improved MSI detection.

Additionally, deep learning algorithms are capable of predicting prognosis in GI cancer by estimating patient survival outcome. Bychkov et al. used the VGG16 model trained with H&E slides only to predict the 5-year disease-specific survival of patients with CRC into either low- or high-risk groups [123]. Their results demonstrated the superior performance of DL-based algorithms compared to manual visual inspection in the stratification of low- and high-risk patients, with a hazard ratio of 2.3 (95% CI: 1.79–3.03). Moreover, the use of different IHC markers to assess the risk for cancer-specific death was demonstrated by Meier et al. Tissue slides were stained with a panel of markers, including Ki67, CD8, CD20 and CD68, and were used as input to train GoogLeNet. Their results showed that a combination of these markers can support the prediction of 5-year survival classification in patients (Ki67 and CD20 vs. CD20 and CD68; hazard ratio: 1.47 vs. 1.33). These works demonstrate the potential of using DL-based algorithms in a clinical setting by adding value to field-of-precision oncology [124].

### 5.2. Weakly Supervised Learning

Weakly supervised learning refers to the use of data labelled with a noisy or imprecise source to predict outcome. Due to increasing challenges associated with algorithm development in computational pathology, such conditions will continue to increase the burden of production and access to larger datasets with time-consuming annotation for model training. To maximize algorithm learning ability from data from limited annotations, multi-instance learning strategies are widely adopted in weakly supervised learning algorithms, especially in the field of computation pathology [125]. The general assumption of a multi-instance learning strategy is that the same histopathological information exists between a WSI and all its patch images. For instance, in a WSI with MSI, all generated tiled images are labelled as “with MSI” [126].

By implying such a straightforward assumption, considerable success has been achieved with a variety of techniques, such as recalibrated multi-instance learning [127] for classification of cancer and dysplasia in GC, graph convolutional network-based multi-instance learning [37] for identification of the presence of tumours in gastric biopsies, S3TA multi-instance learning method [38] for prediction of the presence of epithelial cell nuclei in CRC and GastroMIL [39] for GC diagnosis and risk prediction. Other than GI cancer, multi-instance learning also shows its strong adaptability for cancer tissue classification, such as dual-stream multiple instance learning network [40] and TransMIL [128] for recognition of tumours in lung or breast cancer and cluster-to-conquer [129] for identification of celiac cancer. It is anticipated that multi-instance learning will remain popular for the classification of GI cancer. Beyond the widely used multi-instance learning strategy, other novel learning strategies, network structures and loss function are proposed to perform classification tasks in GI cancer. Klein et al. [130] proposed the use of a VGG architectural active learning network to detect patients with *H. pylori* and obtained an outstanding AUC (>0.80). Li et al. [131] proposed a definition of a new hybrid supervised learning method combining pixel-level and image-level annotations and achieved exceptional performance (sensitivity, 1.0000; AUC, 0.9906) for the recognition of GC using the ResNet-34 and Otsu methods. Kwabena et al. [132] proposed the use of dual horizontal squash capsule networks for the identification of malignant and benign regions in CRC and obtained a high AUC (0.998). Chen et al. [133] proposed a new loss function, namely rectified cross-entropy and upper transition loss for the prediction of CRC tumour grade and obtained an average accuracy of 0.76. The study showed that the new learning strategy and network structure were more effective than the optimization of loss function in weakly supervised learning. Furthermore, reduced performance would occur when applying the trained model to multiple datasets, especially for weakly supervised learning algorithms. To overcome such challenges, colour normalization and adversarial learning [134] can be applied to train weakly supervised algorithms from different datasets for cancer grading.

**Table 2 cancers-14-03780-t002:** Histopathologically related deep learning models used for clinical applications in GI cancers. Deep learning algorithm and models are grouped according to their specific computational task and GI cancer type to compare their performance and clinical applications. The sources of the datasets and sample sizes are also summarized.

Author	Degree of Supervision	Task	Cancer Type	Type of WSI	Dataset	Algorithm/ Model	Performance	Clinical Application
Shen et al. [112]	Fully supervised	Classification	Gastric cancer	H&E	Training, validation and testing: 432 WSIs (TCGA-STAD cohort) + 460 WSIs (TCGA-COAD) + 171 WSIs (TCGA-READ) + 400 WSIs (Camelyon16)	DenseNet + Deformable Conditional Random Field model	Accuracy: 0.9398 (TCGA-STAD), 0.9337 (TCGA-COAD), 0.9294 (TCGA-READ), 0.9468 (Camelyon16)	Identification of suspected cancer area from histological imaging
Song et al. [30]	Fully supervised	Classification	Gastric cancer	H&E	Training: 2123 WSIs Validation: 300 WSIs Internal testing: 100 WSIs External validation: 3212 WSIs (daily gastric dataset) + 595 WSIs (PUMCH) + 987 WSIs (CHCAMS and Peking Union Medical College)	DeepLab v3	Malignant vs. benign training AUC: 0.923 Internal testing AUC: 0.931 AUC: 0.995 (daily gastric dataset) AUC: 0.990 (PUMCH) AUC: 0.996 (CHCAMS and Peking Union Medical College)	Diagnosis of gastric cancer
Su et al. [28]	Fully supervised	Classification and detection	Gastric cancer	H&E	Training: 348 WSIs Testing: 88 WSIs External Validation: 31 WSIs	ResNet-18	Poorly differentiated adenocarcinoma vs. well-differentiated adenocarcinoma and other normal tissue F1 score: 0.8615 Well-differentiated adenocarcinoma vs. poorly differentiated adenocarcinoma and other normal tissue F1 score: 0.8977 Patients with MSI vs. without MSI Accuracy: 0.7727 (95% CI 0.6857–0.8636)	Differentiation of cancer grade and diagnosis of MSI
Song et al. [29]	Fully supervised	Classification	Colorectal cancer	H&E	Training: 177 WSIs Validation: 40 WSIs Internal test: 194 WSIs External validation: 168 WSIs	Deep Lab v2 with ResNet34	Adenomatous vs. normal AUC: 0.92 Accuracy: 0.904	Diagnosis of colorectal adenomas
Sirinukunwattana et al. [114]	Fully supervised	Classification	Colorectal cancer	H&E	Training: 510 WSIs External validation: 431 WSIs (TCGA cohort) + 265 WSIs (GRAMPIAN cohort)	Inception V3	Colorectal cancer consensus molecular subtypes 1 vs. 2 vs. 3 vs. 4 Training average accuracy: 70% Training AUC: 0.9 External validation accuracy: 0.64 (TCGA cohort) + 0.72 (GRAMPIAN cohort)External validation AUC:0.84 (TCGA cohort) + 0.85 (GRAMPIAN cohort)	Prediction of colorectal cancer molecular subtype
Popovici et al. [115]	Fully supervised	Classification	Colorectal cancer	H&E	Training: 100 WSIs Test: 200 WSIs	VGG-F	Molecular subtype A vs. B vs. C vs. D vs. EOverall accuracy: 0.84 (95% CI: 0.79−0.88)Overall recall: 0.85 (95% CI: 0.80−0.89)Overall precision: 0.84 (95% CI: 0.80−0.88)	Prediction of colorectal cancer molecular subtype
Korbar et al. [116]	Fully supervised	Classification	Colorectal cancer	H&E	Training: 458 WSIs Testing: 239 WSIs	ResNet-152	Hyperplastic polyp vs. sessile serrated polyp vs. traditional serrated adenoma vs. tubular adenoma vs. tubulovillous/villous adenoma vs. normal Accuracy: 0.930 (95% CI: 0.890−0.959) Precision: 0.897 (95% CI: 0.852−0.932) Recall: 0.883 (95% CI: 0.836−0.921) F1 score: 0.888 (95% CI: 0.841−0.925)	Characterization of colorectal polyps
Wei et al. [118]	Fully supervised	Classification	Colorectal cancer	H&E	Training: 326 WSIs Validation: 25 WSIs Internet test: 157 WSIs External validation: 238 WSIs	Ensemble ResNet×5	Hyperplastic polyp vs. sessile serrated adenoma vs. tubular adenoma vs. tubulovillous or villous adenoma. Internal test mean accuracy: 0.935 (95% CI: 0.896–0.974) External validation mean accuracy: 0.870 (95% CI: 0.827–0.913)	Colorectal polyp classification
Gupta et al. [119]	Fully supervised	Classification	Colorectal cancer	H&E	Training and testing: 303,012 normal WSI patches and approximately 1,000,000 abnormal WSI patches	Customized Inception-ResNet-v2 Type 5 (IR-v2 Type 5) model.	Abnormal region vs. normal region F-score: 0.99 AUC: 0.99	Identification of suspected cancer area from histological imaging
Kather et al. [117]	Fully supervised	Classification and prognosis	Colorectal cancer	H&E	Training: 86 WSIs Testing: 25 WSIs External validation: 862 WSIs (TCGA cohort) + 409 WSIs (DACHS cohort)	VGG19	Adipose tissue vs. background vs. lymphocytes vs. mucus vs. smooth muscle vs. normal colon mucosa vs. cancer-associated stroma vs. colorectal adenocarcinoma epithelium Internal testing Overall Accuracy: 0.99 External testing Overall accuracy: 0.943 High deep stroma score predicts shorter survival Hazard ratio: 1.99 (95% CI: 1.27–3.12)	Colorectal cancer detection and prediction of patient survival outcome
Zhu et al. [31]	Fully supervised	Classification and segmentation	Gastric and colorectal cancer	H&E	Training: 750 WSIs Testing: 250 WSIs	Adversarial CAC-UNet	Malignant region vs. benign region DSC: 0.8749 Recall: 0.9362 Precision: 0.9027 Accuracy: 0.8935	Identification of suspected cancer area from histological imaging
Xu et al. [33]	Fully supervised	Segmentation	Colorectal cancer	H&E	Training: 750 WSIs Testing: 250 WSIs	CoUNet	Malignant region vs. benign region Dice: 0.746 AUC: 0.980	Identification of suspected cancer area from histological imaging
Feng et al. [32]	Fully supervised	Segmentation	Colorectal cancer	H&E	Training: 750 WSIs Testing: 250 WSIs	U-Net-16	Malignant region vs. benign region DSC: 0.7789 AUC:1	Identification of suspected cancer area from histological imaging
Mahendra et al. [120]	Fully supervised	Segmentation	Colorectal cancer	H&E	Training: 270 WSIs (CAMELYON16) + 500 WSIs (CAMELYON17) + 660 WSIs (DigestPath) + 50 WSIs (PAIP) Testing: 129 WSIs (CAMELYON16) + 500 WSIs (CAMELYON17) + 212 WSIs (DigestPath) + 40 WSIs (PAIP)	DenseNet-121 + Inception-ResNet-V2 + DeeplabV3Plus	Malignant region vs. benign region Cohen kappa score: 0.9090 (CAMELYON17) DSC: 0.782 (DigestPath)	Identification of suspected cancer area from histological imaging
Gehrung et al. [34]	Fully supervised	Detection	Oesophageal cancer	H&E and TFF3 pathology slides	Training: 100 + 187 patients Validation: 187 patients External validation: 1519 patients	VGG-16	Patients with Barrett’s oesophagus vs. no Barrett’s oesophagus AUC: 0.88 (95% CI: 0.85–0.91) Sensitivity: 0.7262 (95% CI: 0.6742–0.7821) Specificity: 0.9313 (95% CI: 0.9004–0.9613) Simulated realistic cohort workload reduction: 57% External validation cohort reduction: 57.41%	Detection of Barrett’s oesophagus
Kather et al. [122]	Fully supervised	Detection	Gastric and colorectal cancer	H&E	Training: 81 patients (UMM and NCT tissue bank) + 216 patients (TCGA-STAD) + 278 patients (TCGA-CRC-KR) + 260 patients (TCGA-CRC-DX) + 382 patients (UCEC) External validation: 99 patients (TCGA-STAD) + 109 patients (TCGA-CRC-KR) +100 patients (TCGA-CRC-DX) +110 patients (UCEC) + 185 patients (KCCH)	Resnet18	Patients with MSI vs. no MSI Training AUC: >0.99 (UMM and NCT tissue bank) AUC: 0.81 (CI: 0.69–0.90) (TCGA-STAD) AUC: 0.84 (CI: 0.73–0.91) (TCGA-CRC-KR) AUC: 0.77 (CI: 0.62–0.87) (TCGA-CRC-DX) AUC: 0.75 (CI: 0.63–0.83) (UCEC) AUC: 0.69 (CI: 0.52–0.82) (KCCH)	Detection of MSI
Echle et al. [36]	Fully supervised	Detection	Colorectal cancer	H&E	Training: 6406 WSIs External validation: 771 WSIs	Shufflenet	Colorectal tumour sample with dMMR or MSI vs. no dMMR or MSI Mean AUC: 0.92 AUPRC: 0.93 Specificity: 0.67 Sensitivity: 0.95 External validation AUC without colour normalisation: 0.95 External validation AUC with colour normalisation: 0.96	Detection of MSI
Cao et al. [121]	Fully supervised	Detection	Colorectal cancer	H&E	Training: 429 WSIs External validation: 785 WSIs	ResNet-18	Colorectal cancer patients with MSI vs. no MSI AUC: 0.8848 (95% CI: 0.8185–0.9512) External validation AUC: 0.8504 (95% CI: 0.7591–0.9323)	Detection of MSI
Meier et al. [124]	Fully supervised	Prognosis	Gastric cancer	H&E IHC staining, including CD8, CD20, CD68 and Ki67	Training and testing: 248 patients	GoogLeNet	Risk of the presence of Ki67&CD20 Hazard ratio = 1.47 (95% CI: 1.15–1.89) Risk of the presence of CD20&CD68 Hazard ratio = 1.33 (95% CI: 1.07–1.67)	Cancer prognosis based on various IHC markers to predict patient survival outcome
Bychkov et al. [123]	Fully supervised	Prognosis	Colorectal cancer	H&E	Training: 220 WSIs Validation: 60 WSIs Testing: 140 WSIs	VGG-16	High-risk patients vs. low-risk patients Prediction with small tissue hazard ratio: 2.3 (95% CI: 1.79–3.03)	Survival analysis of colorectal cancer
Wang et al. [127]	Weakly supervised	Classification	Gastric cancer	H&E	Training: 408 WSIs Testing: 200 WSIs	recalibrated multi-instance deep learning	Cancer vs. dysplasia vs. normal Accuracy: 0.865	Diagnosis of gastric cancer
Xu et al. [37]	Weakly supervised	Classification	Gastric cancer	H&E	Training, validation and testing: 185 WSIs (SRS dataset) + 2032 WSIs (Mars dataset)	multiple instance classification framework based on graph convolutional networks	Tumour vs. normal Recall: 0.904 (SRS dataset), 0.9824 (Mars dataset) Precision: 0.9116 (SRS dataset), 0.9826 (Mars dataset) F1-score: 0.9075 (SRS dataset), 0.9824 (Mars dataset)	Diagnosis of gastric cancer
Huang et al. [39]	Weakly supervised	Classification	Gastric cancer	H&E	Training and testing: 2333 WSIs External validation: 175 WSIs	GastroMIL	Gastric cancer vs. normal External validation accuracy: 0.92 GastroMIL risk score associated with patient overall survival Hazard ratio: 2.414	Diagnosis of gastric cancer and prediction of patient survival outcome
Li et al. [131]	Weakly supervised	Classification	Gastric cancer	H&E	Training and testing: 10,894 WSIs	DLA34 + Otsu’s method	Tumour vs. normal Sensitivity: 1.0000 Specificity: 0.8932 AUC: 0.9906	Diagnosis of gastric cancer
Chen et al. [133]	Weakly supervised	Classification	Colorectal cancer	H&E	Training and testing: 400 WSIs	CNN classifier	Normal (including hyperplastic polyp) vs. adenoma vs. adenocarcinoma vs. mucinous adenocarcinoma vs. signet ring cell carcinoma Overall accuracy: 0.76	Prediction of colorectal cancer tumour grade
Ye et al. [38]	Weakly supervised	Classification	Colorectal cancer	H&E	Training and testing: 100 WSIs	Multiple-instance CNN	With epithelial cell nuclei vs. no epithelial cell nuclei Accuracy: 0.936 Precision: 0.922 Recall: 0.960	Detection of colon cancer
Sharma et al. [129]	Weakly supervised	Classification	Gastrointestinal cancer	H&E	Training and testing: 413 WSIs	Cluster-to-Conquer framework	Celiac cancer vs. normal Accuracy: 0.862 Precision: 0.855 Recall: 0.922 F1-score: 0.887	Detection of gastrointestinal cancer
Klein et al. [130]	Weakly supervised	Detection	Gastric cancer	H&E + Giemsa staining	Training: 191 H&E WSIs and 286 Giemsa-stained WSIs Validation: 71 H&E WSIs and 87 Giemsa-stained WSIs External validation: 364 H&E WSIs and 347 Giemsa-stained WSIs	VGG+ + active learning	*H. pylori* vs. no *H. pylori* External validation AUC: 0.81 (H&E) + 0.92 (Giemsa-stained)	Detection of *H. pylori*

WSI = whole-slide imaging; H&E = haematoxylin and eosin; AUC = area under the curve; CI = confidence interval; TCGA = The Cancer Genome Atlas; STAD = stomach adenocarcinoma; DACHS = Darmkrebs: Chancen der Verhütung durch Screening; MSI = microsatellite instability; dMMR = deficient mismatch repair; TFF3 = trefoil factor 3; DSC = Dice similarity coefficient; UMM = University Medical Centre Mannheim, Heidelberg University; NCT = National Centre for Tumour Diseases; CRC = colorectal cancer; PUMCH = Peking Union Medical College Hospital; CHCAMS = Chinese Academy of Medical Sciences; *H. pylori* = Helicobacter pylori; IHC = immunohistochemistry; CNN = convolutional neural network.

## 6. Clinical Insight for Selected AI Applications in Early Diagnosis and Monitoring of the Progression of GI Cancer

The application of AI in GI cancer is an important and rapidly growing area of research. Early diagnosis and monitoring of GI cancer can minimize cancer incidence, lower mortality rates and reduce the cost of care. Effective clinical management is a prerequisite to achieve these goals, and the incorporation of AI in healthcare systems is indispensable. Many AI-based software applications and algorithms have been developed to assist in the detection of GI premalignant lesions. In addition, several investigations have shown impact through advanced AI algorithms for tumour subtyping by providing valuable information that may affect subsequent treatment planning and prediction of patient outcomes. Finally, promising results have been demonstrated by various intelligential WSI analytical neural networks for the screening of GI cancer, ranging from AI monitoring to providing rapid analysis and diagnosis of GI cancer, optimizing the workflow in an AP laboratory. Based on these promising results, the possible practical applications of AI for histopathological analysis will also be highlighted below.

### 6.1. Detection of Early GI Premalignant Lesions

Early detection of precancerous lesions is crucial to identify certain GI cancers, especially in GC and CRC [135], which are associated with high incidence and mortality. As chronic *H. pylori* infection (Figure 3A) is a major risk factor for GC, evaluation of *H. pylori* is critical in gastric biopsies. Identification of these organisms requires examination of the slide under high magnification (400×). Although this bacterium can be easily identified in H&E-stained samples if present in large quantities, identifying cases with scanty numbers of *H. pylori* can be extremely time-consuming. Although ancillary staining (e.g., Warthin–Starry staining, IHC) is possible, these methods require extra cost and time for diagnosis. Deep learning assistance [35,130] can further speed up GC diagnosis and provide alternative solutions to low-resource settings, where ancillary stains are not readily available. Klein et al. [130] designed VGG-based algorithms under an active learning scheme. By utilizing algorithms that are able to proactively learn and train from standard GC histology images with *H. pylori,* the required amount of pixelwise labelling annotation by pathologists can be significantly reduced, eliminating the need for large, annotated image datasets established by pathologists. Therefore, only a small portion of manual annotation will be required to develop AI applications for detection of *H. pylori* in order to self-generate annotation of regions with *H. pylori*.

Unlike GC, differentiating high-risk colorectal polyps from low-risk polyps (e.g., innocuous hyperplastic polyps) is an important task in CRC screening (Figure 3J). As identifying high-risk polyps is based on characterization of specific types of polyps (e.g., sessile serrated polyps) [136] and considerable interobserver variability exists among pathologists [137,138,139], accurate diagnosis of high-risk polyps is needed for effective and early detection of CRC. As all US adults aged 50–75 years old are recommended by the US Preventive Services Task Force [140] to screen for CRC yearly, AI applications [29,116,118] for classification of high-risk colorectal polyps could demonstrate its clinical utility in prioritising slides with higher likelihood of malignancy. Wei et al. [118] demonstrated that their AI algorithms for colorectal polyp detection can further highlight regions of high-risk polyps on WSIs to provide effective, consistent and accurate diagnoses, demonstrating that AI systems, such as computer-aided diagnosis, can possibly be developed to assist pathologists in the interpretation of CRC WSIs. However, as polyp annotations were provided by several pathologists for AI model training, similar errors were found in both pathologist and AI applications with respect to classifying high-risk polyps and low-risk polyps. This suggests that future AI systems for colorectal polyp detection may require additional development with extra manual annotations by different experienced GI pathologists to reduce errors in classification.

Currently, diagnostic procedures still rely heavily on pattern recognition by pathologists to identify regions of interests. Future AI algorithms will guide pathologist by identifying areas of interest by highlighting *H. pylori* and high-risk polyps for confirmation. Given the promising performance of AI algorithms in accurate delineation of premalignant lesion regions, pathologists might integrate computational pathology to adopt AI-based tools in clinical workflows with increased confidence. Eventually, this will reduce the time spent per case and allow for early detection and diagnosis of patients with GI premalignant lesions.

### 6.2. Tumour Subtyping Characterization and Estimation of Patient Outcome

In histopathology, the identification and classification of tumour subtypes is important for personalized medicine [141]. As certain gene mutations and molecular alterations are associated with specific morphological changes [142], deep-learning-based image analysis has the potential to uncover molecular tumour subtyping and build robust classifiers to enhance treatment response in cancer patients. MSI refers to DNA mismatch repair deficiency resulting in accumulation of mutations within short repetitive sequences of DNA (microsatellites). MSI testing is critical for treatment of GI cancer [143], as CRCs with high MSI are associated with poor response to conventional chemotherapy but improved response to immunotherapy, [144,145] leading to improved median overall survival [146]. However, due to the high cost of PCR-based MSI testing [147], many studies have focused on developing deep learning methods based on H&E imaging to provide fast and accurate MSI detection for GC [28,122] and CRC [36,121,122] in countries with limited resources, especially in Latin America and the Caribbean [148]. Due to the lack of exact 3D quantitative MSI information, training deep learning models without accurate information may limit the prognostic performance in GI cancer patients with MSI. Despite imperfect specificity, the increasing amount of MSI related-AI research indicates that AI will be rapidly applied to screen for MSI in GI cancers to lower laboratory operation costs.

Assessment of survival outcomes in GI cancer is another ongoing research area using AI. Many studies have explored the question of whether deep learning applications can be used as prognostic tools for GC [124] (Figure 3C) and CRC [27,117,123] (Figure 3K,L) based on H&E-stained WSIs. Using deep learning, suspected GI tumour tissue (e.g., tissue microarrays [27,123,124] and stroma [117]) can be assessed quantitatively using H&E-stained WSIs to assist pathologists in identifying patients at high risk of mortality for immediate advanced treatment. Furthermore, some pioneering studies [114,115] have attempted to explore the classification of consensus molecular subtype (CMS) classification of CRC and patient survival outcomes using deep learning. With the capacity to identify tumour subtypes and perform risk assessment of cancer patients, AI applications can provide a cost-effective and time-saving solution for clinical decision making. Guided by AI with high reproducibility and objectivity, more clinically relevant information can be generated from WSIs to allow for improved clinical management to enhance the survival of GI cancer patients.

Performing supplementary laboratory tests, including PCR and IHC tests, remains the gold standard to determine the molecular tumour subtype profile of GI cancer. Advancements in AI algorithms allow for application in clinical settings with limited resources to detect specific biomarkers, such as MSI, using commonly available H&E staining images, facilitating early personalized treatment. Additionally, AI application will allow for improved allocation of medical resource by shortening the time required to identify GI cancer patients at high risk of mortality for earlier treatment.

### 6.3. Screening GI Cancer in Daily Clinical Operation

As clinical follow up is essential to reduce cancer mortality [59], effective pathology service must start by providing early detection and accurate cancer diagnosis. The shortage of pathologists continues to worsen globally, resulting in delayed cancer diagnosis and treatment, especially during the recent COVID-19 pandemic health crisis [149,150,151]. The pressure to provide fast and accurate diagnoses, with overloaded histopathology workforces, has forced the transformation of practice from conventional light microscopy to digital pathology with AI analytical applications. Studies have focused on improving clinical workflow by diagnosing various GI cancers, including oesophageal cancer [34] (Figure 3E–H), GC [30,39] (Figure 3C,I), etc. Pertinently, Gehrung et al. [34] established an H&E-based trained deep learning system to identify and triage oesophageal cases with Barrett’s oesophagus (BO) for early detection of oesophageal adenocarcinoma. The AI application resulted in a workload reduction of more than 55% of for BO identification after triage and further demonstrated that the developed AI system can improve operational efficiency and support clinical decision making in a realistic primary care setting. However, the deep learning system developed by Gehrung et al. [34] did not tackle the problem of successful training algorithms requiring large annotated WSI datasets established by pathologists [16]. Campanella et al. [152] demonstrated a weakly supervised system trained by diagnostic data that can be automatically retrieved from pathology reports to triage prostate, skin and breast cancer cases. A workload reduction of more than 65% [152] was observed. As weakly supervised algorithms have proven their value in cancer screening, efforts should be made to develop AI applications to triage and identify GI cancers for daily operation without further overloading the histopathology workforce.

At present, histological cases are reviewed in chronological order in traditional AP laboratories. Clinical needs and the current acceptability of AI have not been well-studied and discussed. One of the ways to demonstrate the clinical usefulness of AI is to measure the potential workload reduction associated with automatic review of benign/disease-free cases. With the utilization of AI for case screening, cases susceptible to GI cancer can be flagged for rapid review, and cases with no potential indication of GI cancer can be semiautomatically reviewed (Figure 3B,D). By measuring the change in pathologist workload in prospective trials [153], the clinical usefulness of AI can be assessed, enabling earlier and improved treatment planning. Furthermore, using slide-level diagnostics in a weakly supervised setting is also recommended for AI algorithm development. This could also alleviate the overwhelming workload issue in AP laboratories by freeing pathologists from the generation of pixel-level annotation for future algorithmic development used in clinical settings.

## 7. Conclusions

In this review, we outlined the potential of AI for the histological diagnosis of GI cancers. The recent increase in demand and complexity of GI cancer diagnosis has prompted the incorporation of digital pathology into the diagnostic workflow. Digital pathology allows pathology information to be acquired, managed and interpreted in a digital environment, opening up opportunities for computational analysis using AI. We presented the applications of the two major AI methodologies, i.e., machine learning and deep learning, for the diagnosis of GI cancers. Various machine learning algorithms show promising sensitivity and specificity in diagnosing GC. In particular, the most popular machine learning algorithm, support vector machine, demonstrates high robustness and generalization ability. Deep learning, which uses highly complex algorithms to simulate the human neural network, is also a powerful tool in digital pathology. We summarized the performance characteristics and limitations of fully supervised and weakly supervised approaches in terms of classification, segmentation, detection and prognostication of GI cancer. We also presented clinical insights with respect to how AI could facilitate the early detection of lesions, tumour subtyping and widespread screening for cancers. Despite the merits offered by AI, the major challenges of algorithm development, such as variation in histological image data, interpathologist variability with respect to interpretations, model transparency and interpretability in clinical settings, have to be overcome. Although there is an urgent need for AI development due to the overwhelming workload of pathologists, it requires pathologists to spend extra time to provide annotations. This dilemma is remains unsolved. From a clinical perspective, solid external validation and quality controls with reference to a large dataset are important in ensuring acceptable standards of AI. This warrants continuous studies on model design by relying on patch/pixel-level annotation, explainability and generalizability of AI algorithms with the inclusion of variability of the multiethnic populations. Through rigid comparison of the current AI model for GI cancer, our studies have highlighted the current challenges associated with developing AI for histological analysis, providing a detailed summary of current AI algorithms developed for GI cancer and spotlighting clinical insights for future development of GI cancer AI algorithms. As AI algorithms continue to advance, we believe that the transparency of such application will also improve. We also believed that the clinical usefulness of AI could be demonstrated though prospective trials. Thus, the adoption of such conceptual AI will be transformed into effective applications of computational pathology for clinical practice.

## Figures and Tables

**Figure 1 cancers-14-03780-f001:**
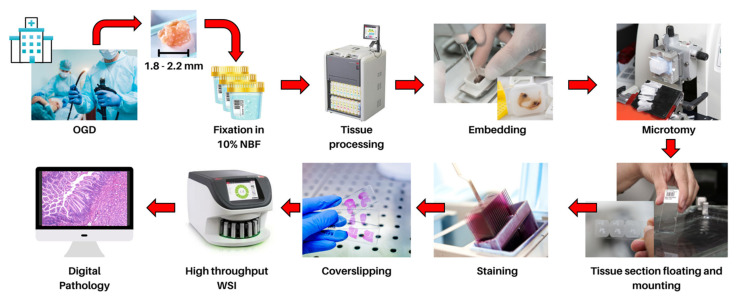
Tissues obtained by endoscopic biopsies are fixed in 10% neutral buffered formalin for 12–24 h. The fixed tissue undergoes dehydration, clearing and impregnation with molten paraffin wax by automatic tissue processors. The tissue is subsequently embedded in a paraffin wax block with proper orientation so tissue sections (3–5 µm thick) can be cut with a microtome. The tissue section is manoeuvred onto a glass slide, stained and mounted with a coverslip to protect and preserve the section. The glass slide is digitized using high-throughput WSI scanners to create a virtual slide to allow for remote diagnosis and large-scale computational pathology.

**Figure 2 cancers-14-03780-f002:**
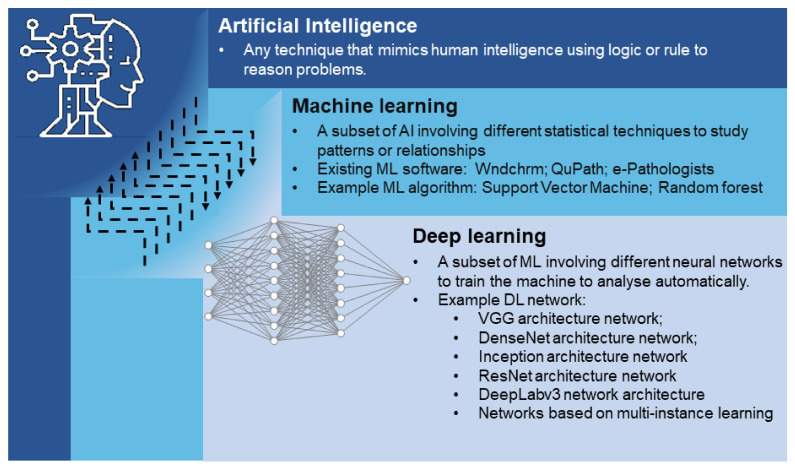
Overview of AI techniques and algorithm development in computational pathology for GI cancers. AI is a concept that mimics human intelligence with respect to learning and problem solving. Deep learning is a subset of machine learning; both are techniques used for the development of AI to study the patterns or relationships in WSIs. Deep learning-based techniques are capable of automatic feature extraction, whereas machine-learning-based techniques require manually designed features. Existing software, algorithms and network architectures discussed in this study are summarized in the figure.

**Figure 3 cancers-14-03780-f003:**
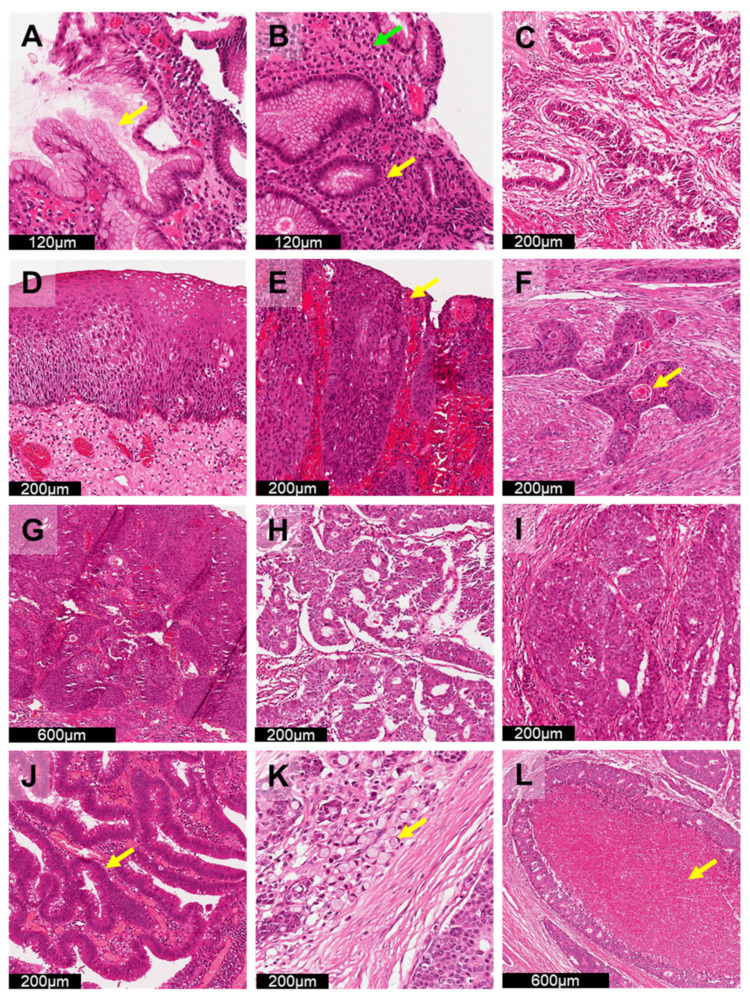
Representative H&E-stained sections of different pathologies along the GI tract. (**A**) Helicobacter-pylori-associated gastritis. Abundant curved rods lining the surface epithelium with underlying mixed inflammatory infiltrate. (**B**) Moderate number of plasma cells (green arrow), indicating chronic inflammation, and neutrophils (yellow arrow), indicating active inflammation. (**C**) Gastric adenocarcinoma. Irregular angulated glands lined by tumour cells with enlarged hyperchromatic nuclei, moderate nuclear pleomorphism and frequent mitotic figures. The background stroma is inflamed and desmoplastic. (**D**) Normal oesophageal squamous epithelium with normal surface maturation. (**E**) Oesophageal squamous epithelium with high-grade dysplasia involving full thickness of the epithelium. The dysplastic cells exhibit enlarged hyperchromatic nuclei, marked nuclear pleomorphism, loss of polarity and lack of surface maturation. No stromal invasion is observed. (**F**) Oesophageal squamous cell carcinoma. Irregular nests of tumour cells infiltrating in desmoplastic stroma. The tumour cells exhibit enlarged hyperchromatic nuclei, marked nuclear pleomorphism and frequent mitotic figures. Squamous pearls (yellow arrow) are observed. (**G**) Oesophageal squamous epithelium with high-grade dysplasia involving full thickness of the epithelium. The dysplastic cells exhibit enlarged hyperchromatic nuclei, marked nuclear pleomorphism, loss of polarity and lack of surface maturation. No stromal invasion is observed. (**H**) Oesophageal adenocarcinoma. Complex cribriform glands lined by tumour cells with enlarged hyperchromatic nuclei, moderate nuclear pleomorphism and frequent mitotic figures. (**I**) Oesophageal adenocarcinoma. Poorly differentiated areas with predominantly solid nests observed. (**J**) Colonic tubular adenoma. Crowded colonic crypts with low-grade dysplasia. The dysplastic cells exhibit pseudostratified, elongated hyperchromatic nuclei. (**K**) Colon adenocarcinoma. Signet ring cells (arrow) with intracellular mucin that displaces the nucleus to the side. (**L**) Colon adenocarcinoma. Cribriform glands with extensive tumour necrosis (arrow).

**Table 1 cancers-14-03780-t001:** Histopathologically related machine learning models used for clinical applications in GI cancers. Machine learning algorithms and models are grouped according to their specific computational task and GI cancer type to compare their performance and clinical application. The sources of the datasets and sample sizes are also summarized.

Author	Task	Cancer Type	Type of WSI	Dataset	Algorithm/ Model	Performance	Clinical Application
Yoshida et al. [24]	Classification	Gastric cancer	H&E	Training and testing: 3062 WSIs	e-Pathologist	Positive for carcinoma or suspicion of carcinoma vs. caution for adenoma or suspicion of a neoplastic lesion vs. negative for a neoplastic lesion Overall concordance rate: 55.6% Kappa coefficient: 0.28 (95% CI: 0.26–0.30) Negative vs. non-negative Sensitivity: 89.5% (95% CI: 87.5–91.4%) Specificity: 50.7% (95% CI: 48.5–52.9%) Positive predictive value: 47.7% (95% CI: 45.4–49.9%) Negative predictive value: 90.6% (95% CI, 88.8–92.2%)	Differentiation and diagnosis gastric cancer grade
Yasuda et al. [25]	Classification	Gastric cancer	H&E	Training and testing: 66 WSIs	wndchrm	Noncancer vs. well-differentiated gastric cancer AUC: 0.99 Noncancer vs. moderately differentiated gastric cancer AUC: 0.98 Noncancer vs. poorly differentiated gastric cancer AUC: 0.99	Differentiation and diagnosis gastric cancer grade
Jiang et al. [106]	Classification and prognosis	Gastric cancer	H&E	Training: 251 patients Internal validation: 248 patients External validation: 287 patients	Support vector machine	Patients might benefit more from postoperative adjuvant chemotherapy vs. patient might not postoperative adjuvant chemotherapy training cohort: 5-year overall survival AUC: 0.796 5-year disease-free survival AUC: 0.805 Internal validation cohort: 5-year overall survival AUC: 0.809 5-year disease-free survival AUC: 0.813 External validation cohort: 5-year overall survival AUC: 0.834 5-year disease-free survival AUC: 0.828	Prognosis of gastric cancer patients and identification of patients who might benefit from adjuvant chemotherapy
Cosatto et al. [26]	Detection	Gastric cancer	H&E	Training set: 8558 patients Test set: 4168 patients	Semi-supervised multi-instance learning framework	Positive vs. negative AUC: 0.96	Detection of gastric cancer
Jiang et al. [27]	Classification	Colon cancer	H&E	Training: 101 patients Internal validation: 67 patients External validation: 47 patients	InceptionResNetV2 + gradient-boosting decision tree machine classifier	High-risk recurrence vs. low-risk recurrence Internal validation hazard ratio: 8.9766 (95% CI: 2.824–28.528) External validation hazard ratio: 10.273 (95% CI: 2.177–48.472) Poor vs. good prognosis groups: Internal validation hazard ratio: 10.687 (95% CI: 2.908–39.272) External validation hazard ratio: 5.033 (95% CI: 1.792–14.132)	Prognosis of stage III colon cancer

WSI = whole-slide imaging; H&E = haematoxylin and eosin; CI= confidence interval; AUC = area under the curve; wndchrm = weighted neighbour distance using compound hierarchy of algorithms representing morphology.

## Data Availability

Data sharing not applicable.

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
