# Peer review of "Current Developments of Artificial Intelligence in Digital Pathology and Its Future Clinical Applications in Gastrointestinal Cancers"

_cancers, 2022, doi:10.3390/cancers14153780_

Round 1

Reviewer 1 Report

This review outlined the potential of AI in the histological diagnosis of GI cancers. The recent increase in demand and complexity of GI cancer diagnosis has prompted the incorporation of digital pathology into the diagnostic workflow.

Digital pathology allows pathology information to be acquired, managed and interpreted in a digital environment, opening up opportunities for computational analysis using AI.

The authors  presented the applications of the two major AI methodologies, machine learning and deep learning, in the diagnosis of GI cancers. Different machine learning algorithms show promising sensitivity and specificity in diagnosing GC.

In particular the most popular machine learning algorithm, the support vector machine, demonstrates high robustness and generalization ability. Deep learning, which uses highly complex algorithms simulating the human neuron network, is also a powerful tool adopted in digital pathology.

They summarized the performance characteristics and limitations of the fully-supervised and weakly-supervised approaches in classification, segmentation, detection and prognostication of GI cancer. They also presented clinical insights on how AI could facilitate the early detection of lesions, tumour subtyping and widespread screening for cancers.

Despite the merits offered by AI, the major challenges of algorithm development, such as colour normalization, interpathologist variability in interpretations, model transparency and interpretability in clinical settings have to be overcome.

From a clinical perspective, solid external validation and quality controls with reference to a large dataset are important in ensuring an acceptable standard of AI. This warrants continuous studies on the explainability and generalizability of AI algorithms with the inclusion of variability of the multi-ethnic population

Author Response

Dear Reviewer 1,

Please see the attachment for our answers to your comments.

Best Regards,

Martin

Reviewer 2 Report

In this work authors talks about Current Developments of Artificial Intelligence in Digital Pathology. I found the methodological part to be well justified and reasonable for this type of analysis. Although the manuscript is overall well-written and structured, it might benefit from additional spell/language checking.

The introduction is deprived of the related work with the recent literature.

What are the key issues present study has addressed?

There are several interesting papers that look into Artificial Intelligence in healthcare. For instance, the below papers has some interesting implications that you could discuss in your Introduction and how it relates to your work. 

Vulli, A.; et al.. Fine-Tuned DenseNet-169 for Breast Cancer Metastasis Prediction Using FastAI and 1-Cycle Policy. Sensors 2022, 22, 2988.

Ali, Farman, et al. "A fuzzy ontology and SVM–based Web content classification system." IEEE Access 5 (2017): 25781-25797.

Srinivasu, Parvathaneni Naga, et al. "Classification of skin disease using deep learning neural networks with MobileNet V2 and LSTM." Sensors 21.8 (2021): 2852.

What are the practical implications of your research? 

Authors should further clarify and elaborate novelty in their contribution.

Conclusion is too short. Add more explanation. 

What are the limitations of the present work?

Author Response

Dear Reviewer 2,

Please see the attachment for our answers to your comments.

Best Regards,

Martin

Reviewer 3 Report

Authors review machine and deep learning approaches used to classifiy and diagnose gasttrointestinal cancers in digital pathology based images of HE stainings.

In machine learning and deep learning methods algorithms are trained / curated. For instance tumor borders are selected and marked on the image Line 274-276 by pathologist. F: Domain-inspired features require the intrinsic domain knowledge of pathologists and oncologists, whilst domain-agnostic features include the general computational features used in the machine learning algorithms. What in my opinion is not addressed sufficiently in this review is the accuray of a pathologist itself. AUC for several algorithms are described, all trained by a pathologist. Is de ÄUC"of a pathologist 1.00? Were on hindsight, diagnoses first missed by a pathologist. What is the AUC required to be able to use the algorithm is dianostics?

The title is somewhat misleading as AI is not used , nor described in GI. Only machine learning and deep learning are described.

One of the benefits stated by the authors is that new biologixcal insight could be aquired. Please give examples of this.

There are many algorithms mentioned, the manuscript would benefit from an overview of these algorithms and a description of what the do and what makes them different from the other algorithms. As it is now, it is a summary of algorithms used and readers would benefit from a brief description of what these algorithms entail.

Author Response

Dear Reviewer 3,

Please see the attachment for our answers to your comments.

Best Regards,

Martin

Round 2

Reviewer 2 Report

.

Reviewer 3 Report

Authors have addressed /  commented to most -not all - of my comments in their respons.